# Design of a Novel Double Negative Metamaterial Absorber Atom for Ku and K Band Applications

**Saif Hannan** [1,*] , **Mohammad Tariqul Islam** [1,*] , **Ahasanul Hoque** [1] , **Mandeep Jit Singh** [1] and **Ali F. Almutairi** [2,*]

1 Centre of Advanced Electronic and Communication Engineering, Faculty of Engineering and Built Environment, Universiti Kebangsaan Malaysia, Bangi 43600, Selangor, Malaysia
2 Electrical Engineering Department, College of Engineering and Petroleum, Kuwait University, Safat 13060, Kuwait
* Correspondence: p98220@siswa.ukm.edu.my (S.H.); tariqul@ukm.edu.my (M.T.I.); ali.almut@ku.edu.kw (A.F.A.); Tel.: +60-135-228-124 (S.H.); +60-193-666-192 (M.T.I.); +965-99-843-420 (A.F.A.)

**Abstract:** This paper presents a multiband metamaterial (MM) absorber based on a novel spiral resonator with continuous, dual, and opposite P-shape. The full wave analysis shows 80.06% to 99.95% absorption at frequencies range for Ku and K bands for several substrate materials of 100 mm$^2$ area. The results indicate that the absorption rate remains similar for different polarizing angles in TEM mode with different substrates. With FR4 (Flame Retardant 4) substrate and 64 mm$^2$ ground plane, the design acts as single negative (SNG) MM absorber in K band resonance frequencies (19.75–21.37 GHz) and acts as double negative (DNG) absorber in Ku band resonance frequencies (15.28–17.07 GHz). However, for Rogers 3035 substrate and 36 mm$^2$ ground plane, it acts as an SNG absorber for Ku band resonance frequency 14.64 GHz with 83.25% absorption and as a DNG absorber for K band frequencies (18.24–16.15 GHz) with 83.69% to 94.43% absorption. With Rogers 4300 substrate and 36 mm$^2$ ground plane, it acts as an SNG absorber for Ku band at 15.04 GHz with 89.77% absorption and as DNG absorber for K band frequencies (22.17–26.88 GHz) with 92.87% to 93.72% absorption. The design was fabricated with all three substrates and showed quite similar results as simulation. In comparison with other broadband absorbers, this proposed MM absorber illustrated broad incidence angles in TEM mode.

**Keywords:** metamaterial absorber; double negative; dual-band

## 1. Introduction

Electromagnetic (EM) absorbers are recent trends in the field of antenna design, sensing, electromagnetic clocking, low cross-section materials for radar, and stealth technologies for military purposes and thermo-photovoltaic applications [1–12]. A perfect metamaterial (MM) absorber [13] is expected to absorb almost the entire EM signal, with very little or none to be reflected back to the source. So, researchers are working hard to design a perfect EM absorber with less scattering and reflection of EM waves from it [14–16]. An absorber is a double negative (DNG), if both permittivity and permeability of it become negative while the EM waves pass through it, as a result, the refractive index will also be negative. If either permittivity or permeability is negative, it acts as a single negative (SNG) absorber, in this case, the refractive index can be either negative or positive.

To increase stealth performances, radar absorbing surfaces are used. Furthermore, polarization angle insensitive properties along with broad band absorption, are the essential objectives of perfect MM absorbers [17,18]. Nowadays, symmetrical structures of EM absorbers are designed to attain polarization-independent EM absorption, like unique geometry of unit cells with circular shapes,

slip-ring- cross resonators and array of these unit cells [19–22]. FR4 substrates are the most common dielectric medium used in these designs. However, the bandwidths of these absorbers are still narrow [11,23,24]. Some works were done with a very small size of unit cells, but they compromised the bandwidth [25–27]. Many researchers are working in this particular field to design a perfect MM absorber with broad band and polarization-insensitive features [28] to employ in stealth and radar systems. Most of the absorber atoms are designed to C and X bands, but absorber for Ku and K bands are rare to find. An absorber for Ku and K bands could be useful for applications like remote sensing, data collection for weather forecasts, wildlife survey, vehicular communication through satellites, etc.

In this paper, a novel spiral resonator with continuous, dual, and opposite P-shape as a unit cell is proposed for almost entire Ku and K band frequencies, which is polarization insensitive for wide incidence angle absorption. The cell was designed on CST microwave studio 2017 software (which was installed on a computer of Intel® Core i3-2120 CPU @ 3.30 GHz and 8.00 GB RAM and Windows 10 operating system), as three layers, patch, a dielectric substrate, and a ground plane. A $10 \times 10$ mm unit cell was considered with patch engraved on the top. The patch has a continuous and flipped P-shape resonator with a square border. The width of the border and the patch wire is the same. An FR4 substrate with $8 \times 8$ mm ground plane was used. The patch and the ground planes are of annealed copper of thickness 0.035 mm. Both normal incidence and oblique incidence of the polarized TEM waves were considered. The cell was replaced later with Rogers RT 3035 and Rogers RT 4003 substrates with the same patch but with a modified ground of $6 \times 6$ mm plane and broad band resonance frequencies in Ku and K bands with more than 80% absorption was obtained. The average time for simulation on CST for the cell to get outputs was around 11 min.

## 2. Design Methodology

The proposed MMA unit cell was designed with FSS (frequency selective surface) patch on three different substrates of different dielectric properties and thickness, backed by a copper ground plane. The mostly used FR4 material was selected as the primary dielectric substrate (dielectric constant $\epsilon_r = 4.6$) with a substrate layer of a thickness of 1.578 mm. The patch and the ground plane are made of copper (annealed and lossy with conductivity $5.8 \times 10^7$ S/m) of thickness 0.035 mm. Figure 1 shows the proposed unit cell with top and back geometry. Here, the metallic FSS layer in Figure 1a (with copper) is presented by a yellow color, and the rest shows the dielectric substrate. Figure 1b shows the ground plane made up of copper (here, for FR4 substrate) just below the substrate. The dimension of the unit cell is shown in Figure 2 and detailed in Table 1. The proposed unit cell has a $10 \times 10$ mm surface. Considering the top surface lying on the x-y plane, the cell was designed and critically analyzed for different incident polarization angles. Incident EM waves were propagated from positive z-direction keeping perfect electric field along the x-axis and perfect magnetic field along the y-axis.

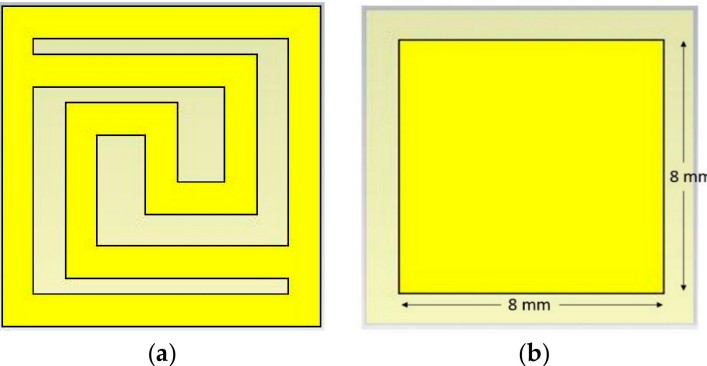

|    (a)    |    (b)    |

**Figure 1.** Design of the proposed unit cell (**a**) patch, (**b**) ground (for FR4 (Flame Retardant 4) substrate).

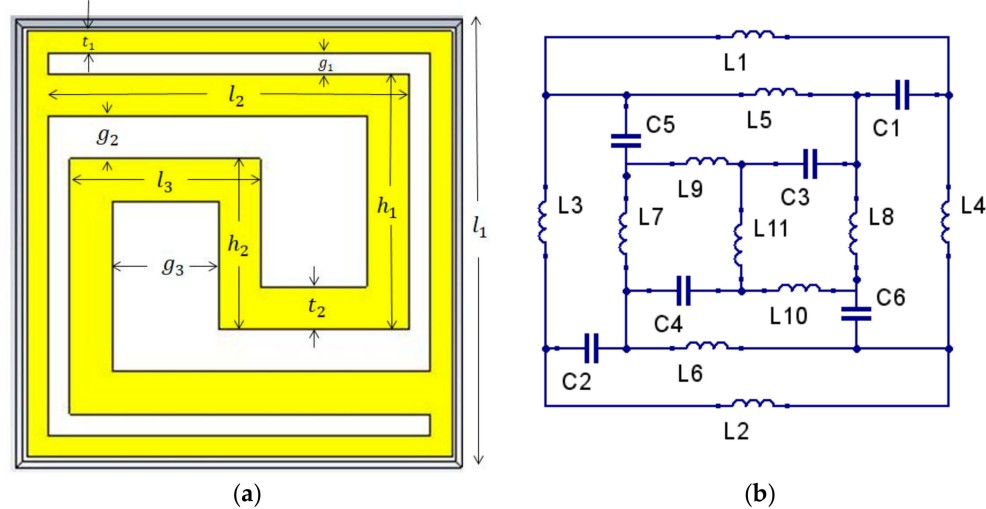

|     |     |
| --- | --- |
| (**a**) | (**b**) |

**Figure 2.** (**a**) Unit cell dimension for patch, (**b**) equivalent circuit.

**Table 1.** Dimension of the continuous, dual, and opposite P-shaped spiral resonator.

| Parameter | $l_1$ | $l_2$ | $l_3$ | $h_1$ | $h_2$ | $t_1$ | $t_2$ | $g_1$ | $g_2$ | $g_3$ |
| --- | --- | --- | --- | --- | --- | --- | --- | --- | --- | --- |
| Size (mm) | 10 | 8.5 | 4.5 | 6 | 4 | 1 | 1 | 0.5 | 1 | 2.5 |

The patch (with annealed copper) was designed in such a way that, it can absorb mostly at the frequency range of Ku and K bands. The continuous double (and opposite) P-shape acts as the resonator, as shown in Figure 2a. The ground (with same copper layer) acts as a reflector so that no transmission takes place through it. As a result, the $S_{11}$ and $S_{21}$ parameters are found (Figure 3) in the required ranges to ensure maximum absorption of the incident frequencies.

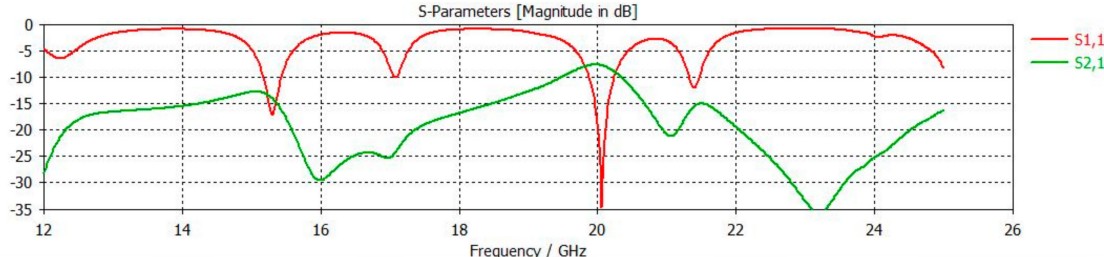

**Figure 3.** S parameters found after simulation for the model (for FR4 substrate).

## 3. Results

We know, the Equation for absorption is

$$A(\omega) = 1 - R(\omega) - T(\omega) \tag{1}$$

where $R(\omega)(= S_{11})$ is reflection coefficient and $T(\omega)(= S_{21})$ is transmission coefficient.

A perfect MMA is understood by restraining the transmitted and reflected EM waves to boost the absorption ratio. If no reflection and transmission take place, the MMA will act as a perfect absorber, as per Equation (1). The proposed design was simulated for Ku and K band frequency range (12–26 GHz) with FR4 substrate. After simulation on CST microwave studio, the values of $S_{11}$ and $S_{21}$ parameters were taken with their absolute values from their real and imaginary values and used them on Matlab with proper commands for absorption, permittivity, permeability, and refractive index in both Nicolson-Ross-Weir (NRW) method and direct refractive index (DRI) method. The following

Equations were used in Matlab commands to get permittivity, permeability, and refractive index (in NRW and DRI methods).

Permittivity (relative),

$$\epsilon_r = \frac{2}{\sqrt{-K_\theta d}} \frac{1 - v_1}{1 + v_1} \tag{2}$$

and permeability (relative),

$$\mu_r = \frac{2}{\sqrt{-K_\theta d}} \frac{1 - v_2}{1 + v_2} \tag{3}$$

where, $v_1 = S_{21} + S_{11}$, $v_2 = S_{21} - S_{11}$, $k_\theta = \frac{\omega}{c}$.
where $\omega = 2\pi f$ ($f$ is the frequency of applied EM wave) and $c$ = speed of light, and $d$ = thickness of the substrate.

Refractive index by Nicolson-Ross-Weir (NRW) method

$$= -\text{real}\left(\eta_r\right) \tag{4}$$

and refractive index by direct refractive index (DRI) method

$$= \text{real}\left(\eta\right) \tag{5}$$

where, $\eta_r = \sqrt{\epsilon_r \mu_r}$, and $\eta = \frac{c}{i\pi f d} \sqrt{\frac{(S_{21}-1)^2-(S_{11})^2}{(S_{21}-1)^2+(S_{11})^2}}$.

Also, the cell was fabricated and tuned to get $S_{11}$ and $S_{21}$ parameters at the resonance frequencies found in simulation, using Matlab to get the absorption, permittivity, and permeability using Equations (1)–(5). The graphs for permittivity, permeability, and absorption rate with respect to frequencies found from Matlab simulation for both simulated and measured data are shown in Figure 4.

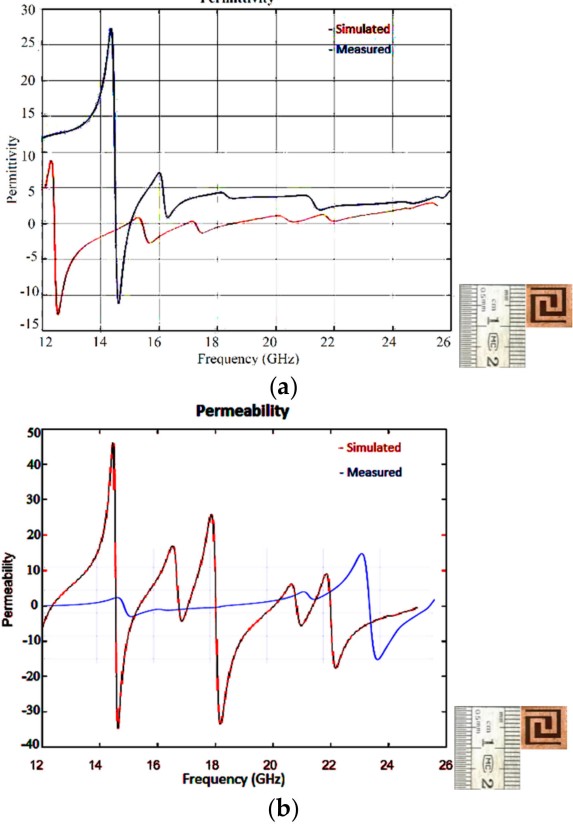

**Figure 4.** *Cont.*

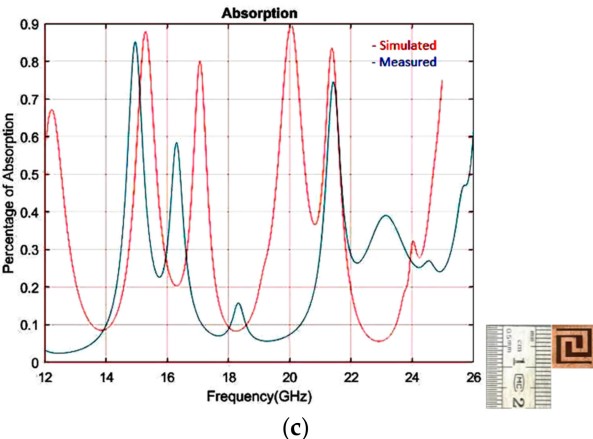

(**c**)

**Figure 4.** Simulated and measured results for FR4 substrate and 8 × 8 ground (**a**) permittivity vs. frequency graph, (**b**) permeability vs. frequency graph, (**c**) percentage of absorption vs. frequency graph.

The maximum absorptions of 98.04% and 88.93% were found with negative permittivity, permeability, and refractive index (both NRW and DRI methods) for 15.3 GHz and 17.04 GHz frequencies respectively with FR4 substrate in Ku band, which is the DNG absorption property shown by design. In the K band region, SNG property was found with a negative value of either permittivity or permeability and negative value of the refractive index by NRW method at 20.06 GHz and 21.3 GHz with 85.93% and 86.68% absorptions respectively, as shown in Table 2 and Figure 4. So, this design acts as a metamaterial absorber at those frequencies shown in Table 2.

**Table 2.** Maximum absorptions with double negative (DNG) and single negative (SNG) values.

| Band | Ku Band (12–18) | | K Band (18–26.5) | |
|---|---|---|---|---|
| Frequency | 15.3 GHz | 17.04 GHz | 20.06 GHz | 21.3 GHz |
| Permittivity | −0.6466 | −0.4076 | −0.1 | 0.8337 |
| Permeability | −1.009 | −0.1276 | 2.252 | −1.352 |
| Refractive Index (NRW) | −0.9062 | −0.4373 | −0.5395 | −0.3514 |
| Refractive Index (DRI) | −0.9062 | −0.4373 | 0.5395 | 0.3514 |
| Absorption | 0.9804 | 0.8893 | 0.8593 | 0.8668 |

Polarization of the incident waves were also considered for the unit cell at different angles (20, 40, 60, and 80 degrees) in TEM mode and found almost same results with insignificant deviations as shown in Table 3. 97.85% to 97.03% absorption were found for DNG properties of EM waves at 15.28 GHz to 15.29 GHz and 80.28% to 89.2% absorption at 16.97 GHz to 17 GHz in Ku band. Whereas in K band, with SNG properties of EM waves, 80.08% to 99.95% absorption was found for 19.75 GHz to 20.02 GHz and 81.41% to 93.52% absorption at 20.23 GHz to 21.37 GHz respectively.

The simulation was done with other substrate materials like, Rogers RT 3035 and Rogers RT 4003 and also fabrication was done using these substrates. It was observed that, with these two substrate materials, we have to change the ground plane significantly to get maximum absorption with DNG properties. The ground was modified with 36 mm$^2$ areas of annealed copper of the same thickness, as shown in Figure 5. The simulative and measured results of permittivity, permeability, and absorption rate against frequency range for RT 3035 (dielectric constant = 3.5, thickness = 0.76 mm) substrate is shown in Figure 6.

**Table 3.** Absorption at different polarizing angles in TEM mode for FR4 substrate.

| Polarization Angle $\varphi$ | Frequency Band | Resonance Frequency (GHz) | Max Absorption (%) | $\in$ | $\mu$ | $\eta$ | EM Mode | Substrate | Dielectric Constant ($\in_r$) |
|---|---|---|---|---|---|---|---|---|---|
| 0 | Ku | 15.3 | 98.04 | −0.6466 | −1.009 | −0.9062 | | | |
| | | 17.04 | 88.93 | −0.4076 | −0.1276 | −0.4373 | | | |
| | K | 20.06 | 85.93 | −0.1 | 2.252 | −0.5395 | | | |
| | | 21.3 | 86.68 | 0.8337 | −1.352 | −0.3514 | | | |
| 40 | Ku | 15.29 | 97.85 | −0.7707 | −0.8825 | −0.8786 | TEM | FR4 | 4.6 |
| | | 17 | 85.34 | −0.2505 | −0.772 | −0.4528 | | | |
| | K | 20.02 | 99.95 | −0.1067 | 0.206 | −0.2 | | | |
| | | 21.24 | 81.76 | −0.1156 | 0.8697 | −0.3296 | | | |
| 80 | Ku | 15.29 | 97.85 | −0.7699 | −0.8829 | −0.8785 | | | |
| | | 16.97 | 80.28 | −0.108 | −1.779 | −0.4846 | | | |
| | K | 19.75 | 80.16 | 0.7407 | −1.907 | −0.3075 | | | |
| | | 21.24 | 81.81 | −0.1162 | 2.261 | −0.3302 | | | |

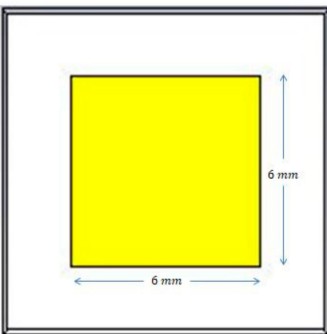

**Figure 5.** Ground plane for Rogers 3035 and Rogers 4003 substrate.

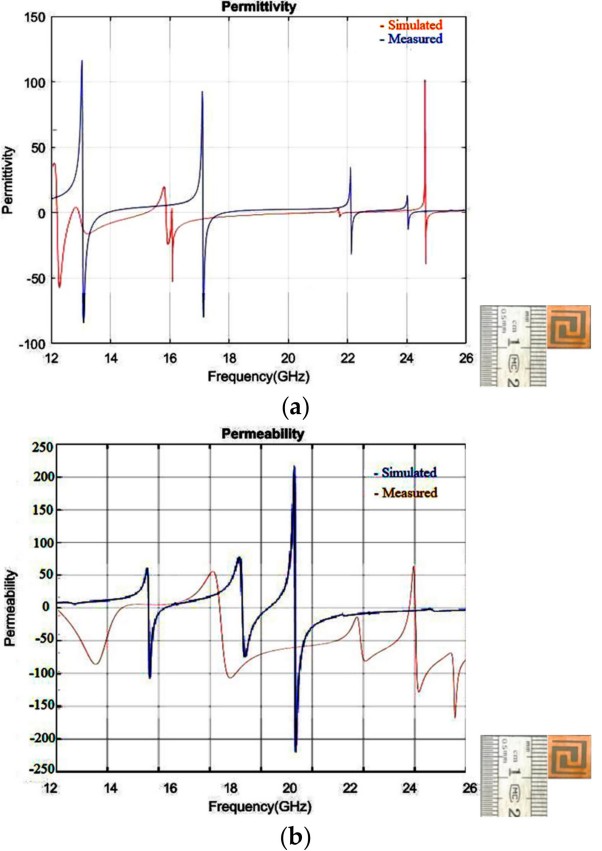

**Figure 6.** *Cont.*

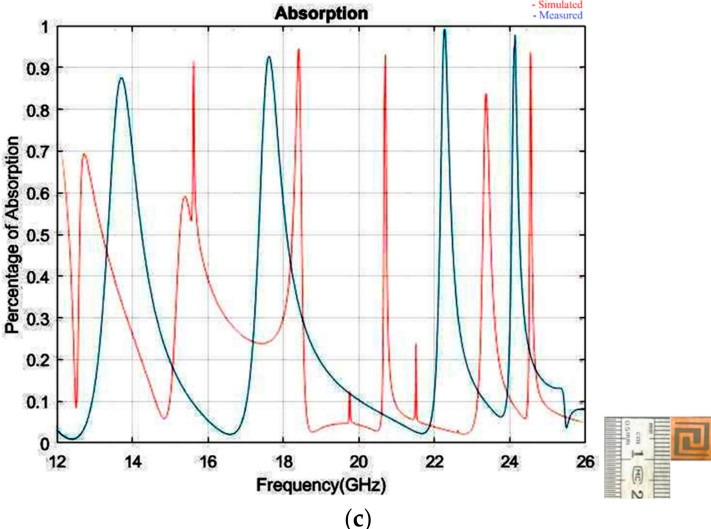

(**c**)

**Figure 6.** Simulated and measured results for Rogers RT 3035 and 6 × 6 ground (**a**) permittivity vs. frequency graph, (**b**) permeability vs. frequency graph, (**c**) percentage of absorption vs. frequency graph.

From Table 4 and Figure 6c, it is seen that 83.25% absorption is found with SNG property at 14.64 GHz for all normal incidence and all polarizing angles in TEM mode of operation in the K band. In the K band region, an excellent absorption was found with DNG property at 18.24, 21.2, 24.62, and 26.15 GHz with 94.43%, 92.92%, 83.72%, and 93.55% absorption respectively. So, this design acts as a perfect absorber with DNG MM characteristics in the K band and SNG MM characteristics in the Ku band with RT 3035 substrate.

**Table 4.** Absorption at different polarizing angles in TEM mode for RT 3035 substrate.

| Polarization Angle $\varphi$ | Frequency Band | Resonance Frequency (GHz) | Max Absorption (%) | $\in$ | $\mu$ | $\eta$ | EM Mode | Substrate | Dielectric Constant ($\in_r$) |
|---|---|---|---|---|---|---|---|---|---|
| 0 | Ku | 14.64 | 83.25 | −6.39 | 5.674 | −2.381 | | | |
| | | 18.24 | 94.43 | −1.195 | −1.064 | −1.224 | | | |
| | K | 21.2 | 92.92 | −2.957 | −9.058 | −4.908 | TEM | Rogers 3035 | 3.5 |
| | | 24.62 | 83.71 | −4.03 | −1.505 | −2.983 | | | |
| | | 26.15 | 93.55 | −1.002 | −3.073 | −1.604 | | | |
| 40 | Ku | 14.64 | 83.26 | −6.39 | 5.674 | −1.383 | | | |
| | | 18.24 | 94.42 | −1.195 | −1.064 | −1.224 | | | |
| | K | 21.2 | 92.92 | −2.957 | −9.058 | −2.4 | | | |
| | | 24.62 | 83.72 | −4.029 | −1.503 | −2.982 | | | |
| | | 26.15 | 93.55 | −1.003 | −3.073 | −1.604 | | | |
| 80 | Ku | 14.64 | 83.25 | −6.39 | 5.674 | −1.382 | | | |
| | | 18.24 | 94.43 | −1.195 | −1.064 | −1.224 | | | |
| | K | 21.2 | 92.92 | −2.957 | −9.058 | −4.908 | | | |
| | | 24.62 | 83.69 | −4.029 | −1.503 | −2.981 | | | |
| | | 26.15 | 93.54 | −1.002 | −2.495 | −1.604 | | | |

With Rogers RT 4003 (dielectric constant = 3.38, thickness = 0.508 mm) substrate, the following results were found, as shown in Figure 7.

With Rogers RT4003 substrate, the results as shown in Figure 7 and Table 5 were achieved. It was observed that the resonance frequencies shifted from those for RT 3035 substrate. Fortunately, the SNG MM characteristics were found in the Ku band and DNG MM characteristics in the K band. 89.78% of absorption was found at 15.04 GHz. 93.28%, 93.72%, and 92.87% absorption were found at 22.17, 25.46, and 26.88 GHz.

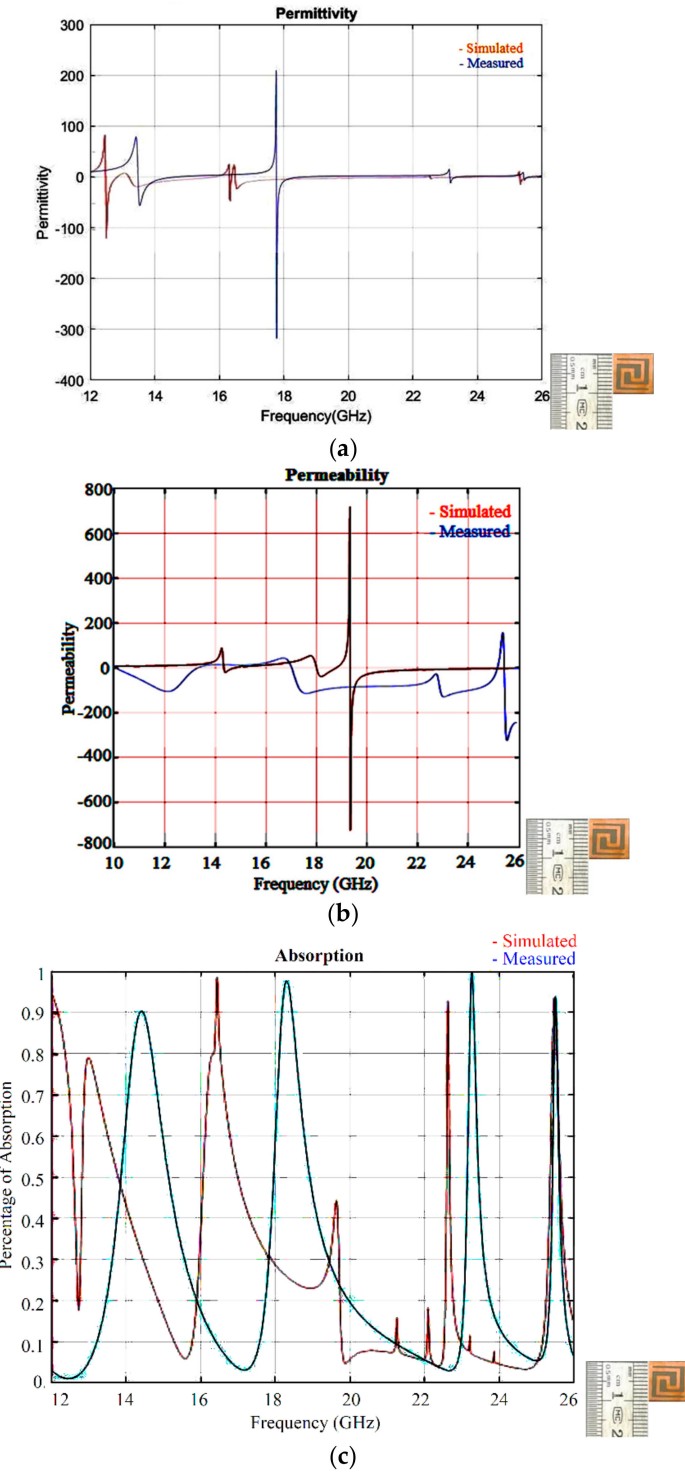

**Figure 7.** Simulated and measured results for Rogers RT 4003 and 6 × 6 ground (**a**) permittivity vs. frequency graph, (**b**) permeability vs. frequency graph, (**c**) percentage of absorption vs. frequency graph.

**Table 5.** Absorption at different polarizing angles in TEM mode for RT 4003 substrate.

| Polarization Angle $\varphi$ | Frequency Band | Resonance Frequency (GHz) | Max Absorption (%) | $\in$ | $\mu$ | $\eta$ | EM Mode | Substrate | Dielectric Constant ($\in_r$) |
|---|---|---|---|---|---|---|---|---|---|
| 0 | Ku | 15.04 | 89.77 | −5.668 | 8.465 | −2.459 | | | |
| | | 22.17 | 93.28 | −1.733 | −7.304 | −4.914 | | | |
| | K | 25.46 | 93.72 | −2.896 | −1.918 | −2.659 | | | |
| | | 26.88 | 92.87 | −0.7353 | −2.112 | −1.481 | TEM | Rogers 4003 | 3.38 |
| 40 | Ku | 15.04 | 89.78 | −5.664 | 8.462 | −2.461 | | | |
| | | 22.17 | 93.28 | −1.733 | −7.304 | −4.914 | | | |
| | K | 25.46 | 93.72 | −2.896 | −2.078 | −2.659 | | | |
| | | 26.88 | 92.87 | −0.7353 | −2.112 | −1.481 | | | |
| 80 | Ku | 15.04 | 89.78 | −5.663 | 8.462 | −2.461 | | | |
| | | 22.17 | 93.28 | −1.733 | −7.304 | −4.914 | | | |
| | K | 25.46 | 93.72 | −2.896 | −2.078 | −2.659 | | | |
| | | 26.88 | 92.87 | −0.7353 | −2.112 | −1.481 | | | |

## 4. Discussion

### 4.1. E-Field, H-Field and Surface Current Analysis

The unit cell was designed with three different substrates (FR4, Rogers RT 3035, and Rogers RT 4003) and as a result, three different types of responses were achieved for the individual substrates from the CST simulations. With FR4 substrate, the E-field, H-field, and surface current distribution were found, as shown in Figure 8. The electric and magnetic field is intense in the regions of bending of the transmission lines inside the patch at 15.3 GHz frequency with 98.04% absorption by it. The values of the permittivity, permeability, and refractive index negative were taken for this case. This is because of the surface current flow in those regions controlled by the transmission lines and the ground placed at the back of the cell. At 17.04 GHz, the electric field becomes more intense in the patch circumference, but magnetic fields shifted towards the center of the patch arms, and hence the value of permeability having much higher value and surface current is less intense than the case for 15.3 GHz, but remains negative with the refractive index and absorption rate reduced to 88.93%. At 20.06 GHz, the electric field is intense at the two opposite arms of the patch only and magnetic field shifts toward the center of the patch, this is because of the surface current distribution is intense at the center and the two opposite arms of the patch. So, permittivity is slightly negative, but permeability becomes positive, and as a result, the refractive index becomes positive by DRI method. However, 85.93% absorption was attained at this frequency. Similarly, at 21.3 GHz, the entire inner circumference of the patch has an intense electric field and associated magnetic field. As a result, the surface current increased significantly, but the electric field became positive, whereas the magnetic field is slightly negative with positive refractive index, and absorption becomes 86.68%. So, with FR4 substrate, the cell acts as a DNG absorber in the Ku band and an SNG absorber in the K band.

For Rogers 3035 substrate, the unit cell shows resonance at 14.64, 18.24, 21.2, 24.62, and 26.15 GHz with 83.25%, 94.43%, 92.92%, 83.71%, and 93.55% absorptions, respectively. The absorptions are high because of the current distributions shown in Figure 9a. The current densities are high around and within the resonator and the ground placed on the opposite side of the substrate. This can be explained with the help of the electric field and magnetic distributions shown in Figure 9b,c.

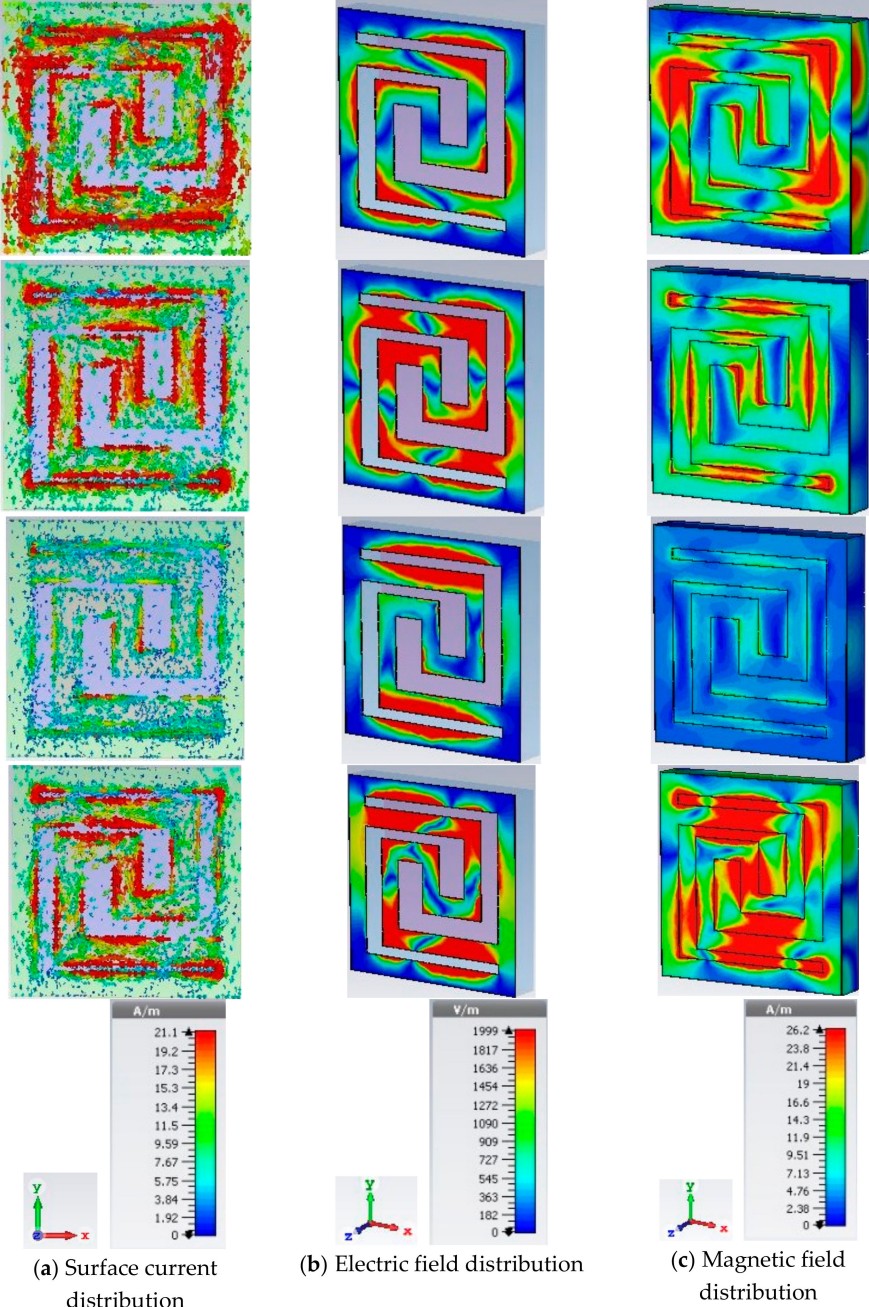

(**a**) Surface current distribution    (**b**) Electric field distribution    (**c**) Magnetic field distribution

**Figure 8.** Instantaneous distribution of (**a**) surface current, (**b**) electric field, and (**c**) magnetic field at 15.3, 17.4, 20.06, and 21.3 GHz, respectively, for FR4 substrate.

At 14.64 GHz, the electric field distribution shows intense resonance at the circumference of the resonator, but the magnetic field shows resonance on the places where electric fields were less intense. This is why permittivity becomes negative, but permeability became positive. Hence in this frequency, the unit cell acts as an SNG absorber. For the rest of the resonance frequencies, the electric and magnetic fields are intense at the same places of the resonator, and hence both permittivity and permeability are negative, resulting the unit cell to act as a DNG absorber.

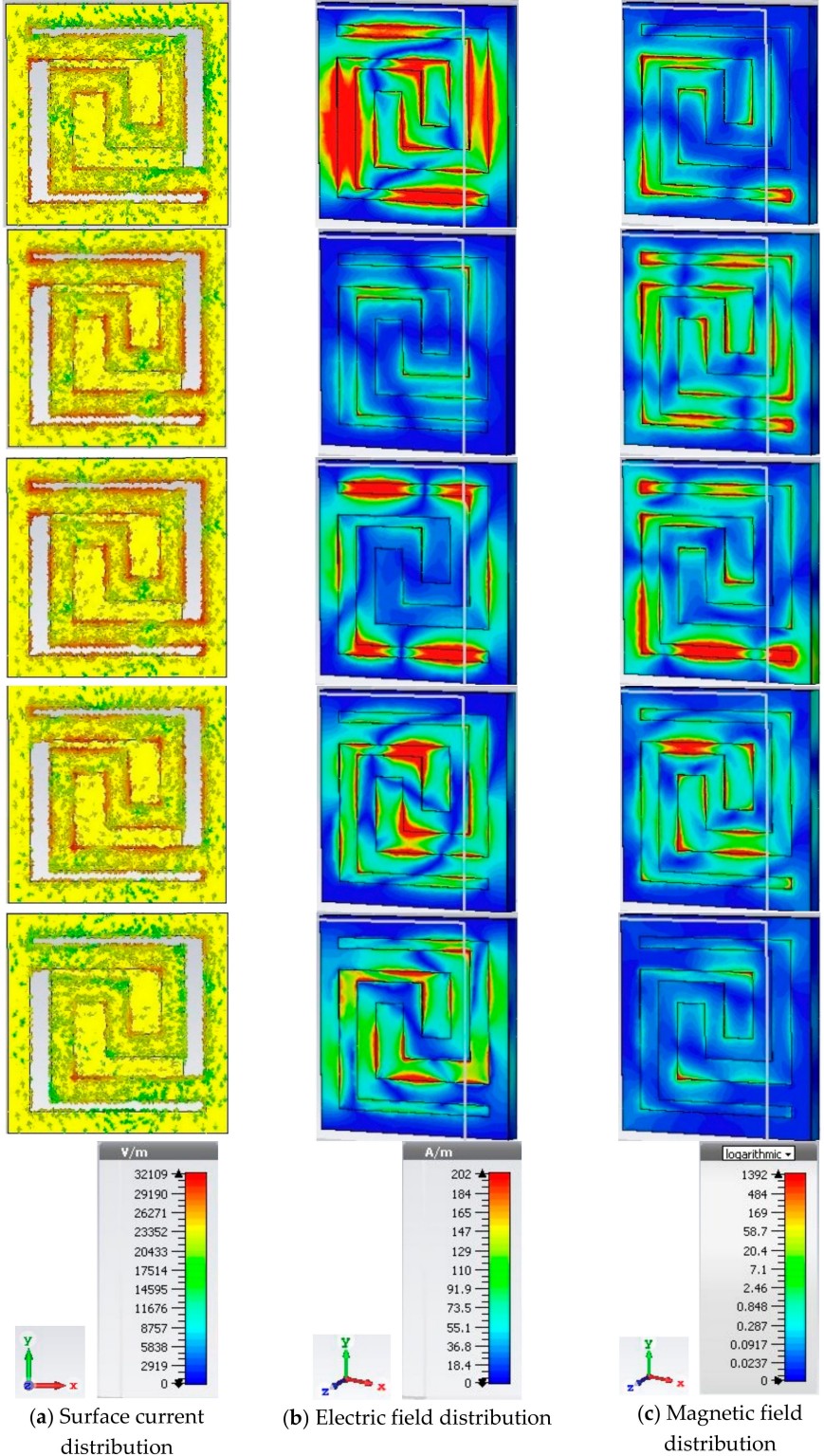

(**a**) Surface current distribution　　(**b**) Electric field distribution　　(**c**) Magnetic field distribution

**Figure 9.** Instantaneous distribution of (**a**) surface current, (**b**) electric field, and (**c**) magnetic field at 14.64, 18.24, 21.2, 24.62, and 26.15 GHz, respectively, for Rogers 3035 substrate.

The unit cell with Rogers 4003 substrate showed resonance at 15.04, 22.17, 25.46, and 26.88 GHz frequencies with 89.77%, 93.28%, 93.72%, and 92.87% absorptions respectively. The cell acts as an SNG absorber at the Ku band (15.04 GHz) and a DNG absorber at the K band. The reason can be explained by Figure 10. For 15.04 GHz, the observed electric field and magnetic fields are not showing agitation

on the same spots of the resonator, the electric field is intense on the top and right-hand side of the resonator, whereas the magnetic field is intense on the bottom and left-hand side of the resonator. Hence both of them have a high but negative value for permittivity and positive value for permeability, as a consequence of the surface current distribution pattern shown for 15.04 GHz.

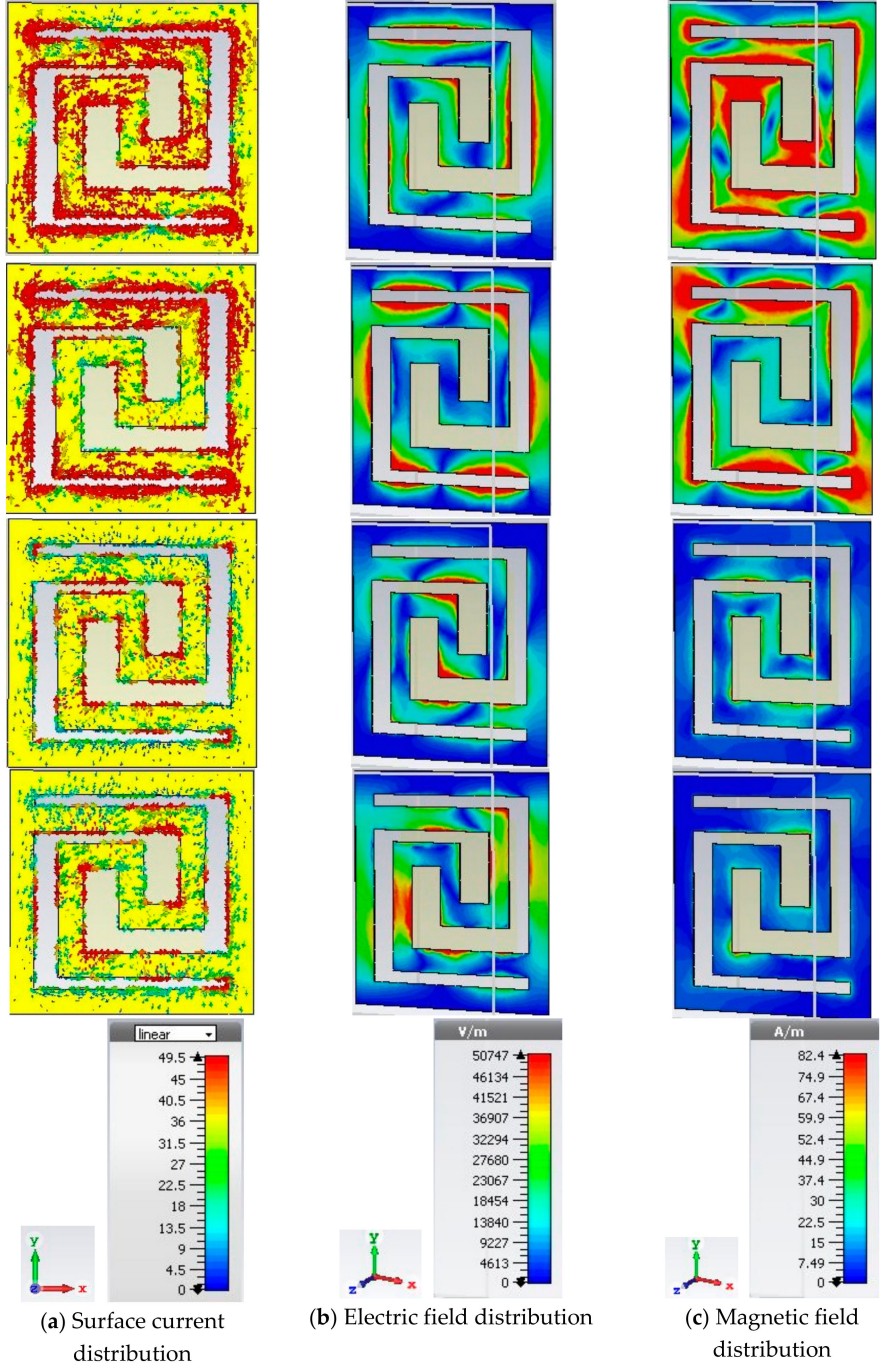

(**a**) Surface current distribution

(**b**) Electric field distribution

(**c**) Magnetic field distribution

**Figure 10.** Instantaneous distribution of (**a**) surface current, (**b**) electric field, and (**c**) magnetic field at 15.04, 22.17, 25.46, and 26.88 GHz, respectively, for Rogers 4003 substrate.

For the rest of the resonance frequencies, the electric and magnetic field distributions show the same expected patterns because of the surface current distributions. We know that the second Maxwell's Equation relates current density with the magnetic and electric field as

$$\nabla \times \boldsymbol{H} = \boldsymbol{J} + \epsilon \frac{\partial \boldsymbol{E}}{\partial t} \tag{6}$$

Moreover, the relation between the current density and the electric field is

$$\boldsymbol{J} = \sigma \boldsymbol{E} \tag{7}$$

It is evident from Equation (6) that, J depends on E, which is also proved on the Figures of electric fields below, where the high density of electric field relates more current densities on surface current distribution Figures. As J and E are changing with respect to time, Magnetic field also changes along the direction normal to E and J. Hence, Magnetic field is less intense on the areas of the resonator where E is intense. This is because E is maximum where J is maximum, but H changes at right angles to E, so it lags behind E, as a result, shows less intense field on those places. This explanation holds for Figures 8–10 also.

### 4.2. Absorption Performance Analysis

The unit cell was designed in such a way that, it can be considered for any polarizing angle of the incident EM wave. The resonator was devised of two oppositely connected, continuous P-shape Copper plates connected with the outer square ring, so that it can create essential capacitance and inductances on the resonator with complete control over performance. This unique architecture helped to attain the desired level of absorption for multiple bands.

The Equation of the absorption of EM wave is given by Equation (1). So, the less is reflection and transmission coefficient, the greater is the absorption. Although we have derived the absorption from S parameters found from simulation results, the phenomena can also be explained from physical observation of the resonator from Figures 8–10.

We know from transmission line theory that, reflection coefficient $R(\omega) = \Gamma = \frac{Z_L - Z_o}{Z_L + Z_o}$ and transmission coefficient $T(\omega) = \tau = \frac{2Z_L}{Z_o + Z_L}$, where $Z_L (= R_L + jX_L)$ is the load impedance and $Z_O$ is the characteristic impedance of the transmission line. The equivalent circuit (Figure 2b) of the resonator shows the load impedance $Z_L \approx 51.99 \ to \ 74.56\Omega$ (calculated) depending on different resonance frequencies and characteristic impedance $Z_O = \sqrt{\frac{L}{C}} \approx 50\Omega$ (this value is predetermined before designing the unit cell and starting the simulation). As transmission coefficient is very small (near to zero due to the ground used (copper is used as ground, hence no transmission took place and EM waves reflected only), it was neglected. So, from Equation (3) it is assumable that, absorption became much higher (more than 80%) due to less reflected wave. Hence the unit cell acted as a perfect absorber at the resonance frequencies.

### 4.3. Comparison of the Unit Cell with Published Works

The performance of this designed unit cell was compared with some relevant works, which are shown in Table 6, below.

The proposed absorber has maximum absorption and a wide range of resonance frequencies with the smallest area of the unit cell compared to others. Moreover, it has a versatile performance probability with different substrates, which is rare in other works.

**Table 6.** Comparison of developed multiband metamaterial (MM) absorber with relevant other papers.

| Ref. # | Year | Size (mm) [Unit Cell] | Substrate Material | Used Frequency Bands | Max Absorption | Application | Resonance Frequency (GHz) |
|---|---|---|---|---|---|---|---|
| Sim et. al. [29] | 2017 | 16.8 × 16.8 | FR4 | X and Ku | >80% | Not specified | 11, 12, 13, 14, 15 |
| Madhav et. al. [30] | 2018 | 40 × 40 | FR4 | Ku, K, and Ka | Not shown | Not specified | 1.9, 7.3, 17.8, 25 |
| Khan et. al. [31] | 2018 | 10 × 10 | RO4350B | X and Ku | Not shown | Hollow waveguide filter and perfect absorber | 7.82, 9.65 |
| Agrawal et. al. [32] | 2018 | 18 × 18 | FR4 | X and Ku | 99.9% | Not specified | 7.6, 8.9, 12.3, 12.8 |
| Jafari et. al. [33] | 2019 | 24 × 24 | FR4 | X and Ku | >84% | Not specified | 8.6, 10.2, 11.95 |
| Our proposed work | 2019 | 10 × 10 | FR4RT 3035RT 4003 | Ku and K | 99.95% | Perfect Absorber | 14.64–15.3, 17.04–18.24, 20.06–21.3, 24.62–26.88 |

## 5. Conclusions

A unique MM absorber was proposed covering the satisfied level of absorption in the Ku and K band region for wide incidence angle EM waves. The design has a spiral resonator with continuous, dual, and opposite P-shape copper patch backed up with a copper ground, to ensure its polarization independence with the horizontally and vertically symmetric structure. The gaps between the spiral arms are also symmetric to keep incidence EM wave angle insensitive. CST 2017 software was used to simulate the design and extract S-parameters to find out absorption, permittivity, permeability, and refractive index by NRW and DRI methods and the cell was fabricated to practically measure these parameters. The measured values are slightly different from simulated values as expected, but still shows absorptions with some negative values of permittivity and permeability. We tried three different substrates for the same design from 10 to 28 GHz and found high absorption rates in Ku and K band regions, with double negative and single negative MM properties. Table 6 shows that our design is much better in comparison with recent works in terms of size of the unit cell, the versatility of using different substrates with the same design, rate of absorption, application, and several resonance frequencies for the highest absorption rate. In all these terms, this design is unique and proves it as a much better absorber, which can be a good candidate for invisibility cloaking, filters, and antennas for satellite communications in Ku and K bands.

**Author Contributions:** S.H. made significant contributions to this study regarding conception, design and analysis, measurement and writing the manuscript. M.T.I. participated in revising the article critically for remarkable intellectual contents and supervised the whole study. A.H. helped in measurement and A.F.A. and M.J.S. revised the manuscript and provided intellectual suggestions.

**Funding:** This work is supported by Universiti Kebangsaan Malaysia research grant.

**Conflicts of Interest:** The authors declare no conflict of interest.

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
