# Peer review of "Design of a Novel Double Negative Metamaterial Absorber Atom for Ku and K Band Applications"

_electronics, doi:10.3390/electronics8080853_

Round 1

Reviewer 1 Report

Please provide the motives for implementing such an absorbers. Have the authors examined other realizations as well? Please give some comparisons.

Regarding the numerical simulations, please provide some evidence on the computational burden and the CPU time.

The authors should provide some comparisons regarding the efficiency of their work with other existing absorbers.

Have the authors used any homogenization technique in order to fine tune the unit-cell of their metamaterial surface? Please give some information.

Finally, some measured results and a fabricated prototype would provide extra level of confidence to your work.

The reference list should be expanded. Some indicative works for finding some extra absorbers and compare the proposed one, are the following:

[1] Li, H., L. H. Yuan, B. Zhou, X. P. Shen, Q. Cheng, and T. J. Cui,“Ultrathin multiband gigahertz metamaterial absorbers,”J. Appl.Phys., Vol. 110, 014909, 1–8, 2011.

[2] Landy, N. Y., S. Sajuyigbe, J. J. Mock, D. R. Smith, andW. J. Padilla, “Perfect metamaterial absorber,”Phys. Rev. Lett.,Vol. 100, 207402, 1–4, 2008.

[3] T. M. Kollatou, A. I. Dimitriadis, S. D. Assimonis, N. V. Kantartzis, and C. S. Antonopoulos, “A Family of Ultra-Thin, Polarization-Insensitive, Multi-Band, Highly Absorbing Metamaterial Structures,” Progress in Electromagnetic Research, vol. 136, pp. 579–594, 2013.

[4] Veysi, M.,  M. Kamyab,  J. Moghaddasi,  and A. Jafargholi,“Transmission phase characterizations of metamaterial coversfor antenna application,”Progress In Electromagnetics ResearchLetters, Vol. 21, 49–57, 2011.Cheng, Y. Z., Y. Wang, Y. Nie, R. Z. Gong, X. Xiong, andX. [5] Wang, “Design, fabrication and measurement of a broadbandpolarization-insensitive metamaterial absorber based on lumpedelements,”J. Appl. Phys., Vol. 111, 044902, 1–4, 2012.

[6] Bilotti, F., A. Toscano, K. B. Alici, E. Ozbay, and L. Vegni,“Design of miniaturized narrowband absorbers based on resonant-magnetic inclusions,”IEEE Trans. Electomagn. Compat., Vol. 53,No. 63, 63–72, 2011.

[7] T. M. Kollatou, A. I. Dimitriadis, S. D. Assimonis, N. V. Kantartzis, and C. S. Antonopoulos, “Mulit-Band, Highly Absorbing, Microwave Metamaterial Structures,” Applied Physics A: Materials Science & Processing, vol. 115, no. 2, pp. 555–561, 2014.

[8] Lee, J. and S. Lim, “Bandwidth-enhanced and polarization-insensitive  metamaterial  absorber  using  double  resonance,”Electron. Lett., Vol. 47, 8–9, 2011.

Reviewer 2 Report

It is a very interesting paper, but the following points are inadequate.

1) There are few descriptions of introduction and discussions.

The introduction does not fully explain the needness of this study, and it is difficult to understand the parameters which aucthors noticed.

The discussion is difficult to understand as the theory is not explained.

I requires additional references and detailed explanations.

2) The research should ensure the validity of the results by two comparisons of theory or experiment.

The results of this study should be explained not only by calculation but also by theory or experimental data.

Reviewer 3 Report

This paper presents a multiband metamaterial (MM) absorber based on a novel spiral resonator with continuous dual and opposite p shape. This is a purely numerical work. To increase the content of the manuscript, in my opinion, the Authors should fabricate a prototype of this metamaterial absorber. In particular, it would be very interesting if the Authors could compare measurements with the numerical data.

Reviewer 4 Report

This is an interesting paper and reports a novel double negative metamaterial. However, the following revisions are recommended before the paper can be published to realise the findings effectively.

The paper needs proofreading and the usage of we, I etc. should be avoided. I can see that the authors use we quite a lot which is uncommon in scientific literature.

While the introduction is good, many relevant papers following similar architecture are omitted such as the ones below. These papers should be cited to demonstrate the author's understanding of the said field.

https://doi.org/10.1016/j.apacoust.2019.03.008

https://doi.org/10.1016/j.mtcomm.2018.12.012

3. The methodology needs improvement it is unclear how the results presented in Fig. 8, 9 and 10 were obtained.

Round 2

Reviewer 2 Report

Suggestions were responeded adequately.

Reviewer 3 Report

Accept.